# The Effects of Apigenin-Biosynthesized Ultra-Small Platinum Nanoparticles on the Human Monocytic THP-1 Cell Line

**DOI:** 10.3390/cells8050444

**Published:** 2019-05-10

**Authors:** Sangiliyandi Gurunathan, Muniyandi Jeyaraj, Min-Hee Kang, Jin-Hoi Kim

**Affiliations:** Department of Stem Cell and Regenerative Biotechnology, Konkuk University, Seoul 05029, Korea; muniyandij@yahoo.com (M.J.); pocachippo@gmail.com (M.-H.K.)

**Keywords:** platinum nanoparticles, oxidative stress, genotoxicity, proinflammatory response, cytokines, apoptosis, DNA damage

## Abstract

Generally, platinum nanoparticles (PtNPs) are considered non-toxic; however, toxicity depends on the size, dose, and physico-chemical properties of materials. Owing to unique physico-chemical properties, PtNPs have emerged as a material of interest for several biomedical applications, particularly therapeutics. The adverse effect of PtNPs on the human monocytic cell line (THP-1) is not well-established and remains elusive. Exposure to PtNPs may trigger oxidative stress and eventually lead to inflammation. To further understand the toxicological properties of PtNPs, we studied the effect of biologically synthesized ultra-small PtNPs on cytotoxicity, genotoxicity, and proinflammatory responses in the human monocytic cell line (THP-1). Our observations clearly indicated that PtNPs induce cytotoxicity in a dose-dependent manner by reducing cell viability and proliferation. The cytotoxicity of THP-1 cells correlated with an increase in the leakage of lactate dehydrogenase, generation of reactive oxygen species, and production of malondialdehyde, nitric oxide, and carbonylated proteins. The involvement of mitochondria in cytotoxicity and genotoxicity was confirmed by loss of mitochondrial membrane potential, lower ATP level, and upregulation of proapoptotic and downregulation of antiapoptotic genes. Decreases in the levels of antioxidants such as reduced glutathione (GSH), oxidized glutathione (GSH: GSSG), glutathione peroxidase (GPx), superoxide dismutase (SOD), catalase (CAT), and thioredoxin (TRX) were indicative of oxidative stress. Apoptosis was confirmed with the significant upregulation of key apoptosis-regulating genes. Oxidative DNA damage was confirmed by the increase in the levels of 8-oxodG and 8-oxoG and upregulation of DNA damage and repair genes. Finally, the proinflammatory responses to PtNPs was determined by assessing the levels of multiple cytokines such as interleukin-1β (IL-1β), IL-6, IL-8, tumor necrosis factor-α (TNF-α), granulocyte-macrophage colony-stimulating factor (GM-CSF), and monocyte chemoattractant protein 1 (MCP-1). All the cytokines were significantly upregulated in a dose-dependent manner. Collectively, these observations suggest that THP-1 cells were vulnerable to biologically synthesized ultra-small PtNPs.

## 1. Introduction

Owing to their catalytic and unique surface-related physicochemical properties, including high surface area, platinum nanoparticles (PtNPs) have garnered considerable interest in the automotive sector, chemical industry, and biomedical device manufacturing industry [1,2,3]. In addition, Pt catalysis is used in hydrogenation reactions, oxidation during organic acid production, isomerization, and dehydrogenation, as well as in the oxidation of carbon monoxide [4,5,6]. Application of PtNPs in industry and biomedicine depends on several factors such as their size, shape and elemental composition, electronic surface structure, presence of capping agent, dispersion state, solubility, and cell type [3,7,8]. Several methods have been developed for PtNPs synthesis; however, these methods have several disadvantages, such as unnecessary toxic effects of nanoparticles due to the presence of various potential harmful contaminants or hazardous chemicals, production of high amounts of waste, and uncontrolled size. Among the physical, chemical, and biological methods of synthesis, the biological method is most viable and ecofriendly. For instance, PtNPs are synthesized using bacteria [9], cyanobacteria [10,11], seaweeds [12], fungi [13], plants [14], and honey [15].

Although biological system-assisted synthesis of PtNPs appears to be simple, non-toxic, and cost-effective, it has certain limitations for biomedical applications; for example, the presence of endotoxins and fragments of biological materials with unwanted biological activity necessitate expensive and time-consuming purification procedures, including downstream processing. Furthermore, generation of biocompatible and stable products is laborious. Therefore, new and efficient methods utilizing suitable reducing agents are required to produce nanomaterials of controllable size, shape, stability, catalytic properties, and surface functionalization within a short period [1]. For example, PtNPs encapsulated within apoferritin exhibited high efficiency of cellular uptake and caused lower level of membrane damage. To avoid unnecessary impurities in nanoparticle preparation, Gurunathan and colleagues used various purified biomolecules such as saponin [16], quercetin [17], resveratrol [18], lycopene [19], R-phycoerythrin [20], and luteolin [21] during the synthesis of various nanoparticles, including silver, palladium, graphene, and gold.

Generally, nanoparticles induce toxicity in human cells via oxidative stress, DNA damage, cell cycle arrest, and mitochondrial dysfunction. Biological activity of nanomaterials is dependent on their size, shape, polarity, porosity, and surface chemistry, type of capping agents, surfactants, and organic solvents used for synthesis. In addition, pristine PtNPs or PtNPs coated with cell-compatible materials exhibit less toxicity. PtNPs in the size range of 5–10 nm did not induce cytotoxicity, oxidative stress, and cell death in A549 and HaCaT cells [22]. Similarly, PtNPs synthesized using citrate as the reducing agent, with particle sizes ranging from 5 to 20 nm, showed significant cytocompatibility, and all the internalized particles accumulated in endosomal/lysosomal vesicles [23]. Functionalization of nanomaterials plays a significant role in cellular toxicity. Human peripheral blood monocytes (THP-1) treated with polystyrene nanoparticles functionalized with carbonyl group (PS-COOH) exhibited no cytotoxicity, whereas polystyrene nanoparticles functionalized with PS-NH2 not only inhibited THP-1 proliferation, but also induced apoptosis in THP-1 cells [24].

Traditionally, Pt-group elements (PGEs) are considered non-toxic and are used extensively as catalysts. However, they cause undesired side effects on human health as airborne particulate matter in vehicle exhaust, which accumulates in living organisms and interferes with various biological pathways [25]. The effects of PtNPs on different cell types require extensive investigation. Although PtNPs are generally considered anti-oxidant molecules, they may cause severe problems once the threshold level is reached. Compared to other metal nanoparticles such as silver nanoparticles (NPs), PtNPs show limited toxicity. Several studies have demonstrated either biocompatibility or considerable cytotoxicity following PtNP exposure, depending on the cell model [26,27]. Interestingly, PtNPs have been shown to act as reactive oxygen species (ROS) scavengers in the presence of other stressors [28,29]. However, in some cases, PtNPs elicited negative bioresponses, including activation of cellular stress, as well as DNA damage and genotoxic effects in vitro [30,31,32]. The addition of PtNPs causes developmental delay and decreases heart beat rate in zebrafish [33]. Contradictory reports on the positive and negative responses to PtNPs have prompted additional evaluation of the cellular responses to PtNPs in human cell lines and biocompatibility of biologically synthesized PtNPs. In this context, we attempted to evaluate the cytotoxic, genotoxic, and immunological responses to biologically synthesized PtNPs in human THP-1 cells.

## 2. Materials and Methods

### 2.1. Materials

The THP-1 cell line was obtained from the American Type Culture Collection (ATCC; Manassas, VA, USA). The cells were maintained as described in Section 2.3. All cells were cultured in 75 cm^2^ tissue culture flasks (Sigma-Aldrich, St. Louis, MO, USA) at 37 °C in the presence of 5% CO_2_ and 95% relative humidity. Dried and hydrated hexachloroplantinic acid (H_2_PtCl_6_·6H_2_O) was purchased from Sigma-Aldrich (St. Louis, MO, USA). Penicillin-streptomycin, trypsin-EDTA, Roswell Park Memorial Institute (RPMI)-1640 cell culture medium, fetal calf serum (FCS), and antibiotic-anti-mycotic reagents were obtained from Life Technologies/Gibco (Grand Island, NY, USA). The in vitro toxicology assay kit was purchased from Sigma-Aldrich. The reagent kits for the measurement of malondialdehyde (MDA), protein carbonyl content, and antioxidant assay were purchased from Sigma-Aldrich. All other chemicals were purchased from Sigma-Aldrich unless otherwise stated.

### 2.2. Synthesis and Characterization of PtNPs

PtNPs were synthesized by reducing PtCl_6_ 2- ions into PtNPs by mixing 10 mL of 1 mg/mL apigenin with 90 mL of 1 mM aqueous H_2_PtCl_6_.6H_2_O. The mixture was maintained at 100 °C (on a hotplate) in a sealed flask to avoid evaporation for 1 h, as temperature catalyzes the reduction process. For control experiments, identical amounts of platinum solution and apigenin were maintained separately under the same reaction conditions. The reduced platinum solution was sonicated for 10 min to separate platinum nanomaterials from the biomolecules. After sonication, the solution was filtrated with a 0.2-µm syringe filter. The reduced platinum metal was purified by repeated centrifugation at 5000× *g* for 30 min and the pellets were washed with distilled water to remove the impurities. Scheme 1 indicates the various steps involved in synthesis and purification of the PtNPs. Purified PtNPs were characterized using various analytical techniques such as UV-vis spectroscopy, X-ray diffraction (XRD), Fourier transform infrared spectroscopy (FTIR), dynamic light scattering (DLS), scanning electron microscopy (SEM), and transmission electron microcopy.

### 2.3. Cell Culture Conditions and PtNP Exposures

THP-1 cells were cultured in RPMI-1640 cell culture medium supplemented with 10% FCS, 2 mM L-glutamine, 10 mM HEPES, 1 mM pyruvate, 100 U/mL penicillin, and 0.1 mg/mL streptomycin (Sigma-Aldrich). The cells were sub-cultured usually twice a week with 1 × 10^6^ viable cells/mL and incubated at 37 °C in a 5% CO_2_ atmosphere. The medium was replaced the next day with 100 µL fresh media and the cells were incubated for 24 h prior to PtNP exposure. Experiments were performed in 96-, 24-, and 12-well plates and 100-mm cell culture dishes, as required. Cells were treated with various concentrations of PtNPs or the required dose of PtNPs.

### 2.4. Cell Viability Assay

Cell viability was measured using cell counting kit-8 (CCK-8; CK04-01l; Dojindo Laboratories, Kumamoto, Japan). Briefly, THP-1 cells were plated in 96-well flat-bottom culture plates containing various concentrations of PtNPs. After 24 h culture at 37 °C in a humidified 5% CO_2_ incubator, the CCK-8 solution (10 μL) was added to each well, and the plate was incubated for another 2 h at 37 °C. The absorbance was measured at 450 nm using a microplate reader (Multiskan FC; Thermo Fisher Scientific, Waltham, MA, USA).

### 2.5. BrdU Cell Proliferation Assay

Cell proliferation was determined according to manufacturer’s instructions (Sigma-Aldrich, St. Louis, MO, USA). Cells were incubated with various concentrations of PtNPs for 24 h; the BrdU labeling solution was added to the culture medium 2 h before the end of the incubation. The cells were fixed and the level of incorporated BrdU was determined using the BrdU enzyme-linked immunosorbent assay (ELISA) kit (Roche, Basel, Switzerland) following the manufacturer’s instructions. Proliferation of the untreated cells at 0 h was considered 100%.

### 2.6. Assessment of Membrane Integrity

The membrane integrity of THP-1 cells was evaluated using a lactate dehyrogenase (LDH) cytotoxicity detection kit. Briefly, the cells were exposed to various concentrations of PtNPs for 24 h. Subsequently, 100 μL of cell-free supernatant from each well was transferred in triplicate into the wells of a 96-well plate, and 100 μL of the LDH reaction mixture was added to each well. After 3 h of incubation under standard conditions, the optical density of the final solution was determined at a wavelength of 490 nm using a microplate reader.

### 2.7. Cell Mortality Assay

Cell mortality was evaluated using the trypan blue assay as described previously [34]. THP-1 cells were plated in the wells of 6-well plates (1 × 10^5^ cells per well) and incubated for 24 h with various concentrations of PtNPs. Cells cultured in medium without PtNPs were used as controls. After 24 h, the cells were detached using 300 µL trypsin–EDTA solution, and both adherent and suspended cells were collected. The mixture of the supernatant and detached cells was centrifuged at 1200 rpm for 5 min. The pellet was mixed with 700 µL trypan blue solution and dispersed. After 5 min of staining, the cells were counted using a cytometer. The viable cells were unstained and the dead cells were stained blue. Three independent experiments were performed in triplicate. The mean and standard deviation were calculated. Cell proliferation is expressed as the percentage of viable cells relative to the appropriate control.

### 2.8. Determination of ROS, MDA, Nitric Oxide (NO), and Carbonylated Protein Levels

ROS were estimated as described previously [35]. THP-1 cells were seeded into wells of 24-well plates at a density of 5 × 10^4^ cells per well and cultured for 24 h. After washing twice with phosphate-buffered saline (PBS), fresh medium containing various concentrations of PtNPs was added and incubated for 24 h. The cells were supplemented with 20 μM DCFH2-DA, and incubation was continued for 30 min at 37 °C. The cells were rinsed with PBS, 2 mL PBS was added to each well, and the fluorescence intensity was determined using a Gemini EM spectrofluorometer (Molecular Devices, Sunnyvale, CA, USA) at an excitation wavelength of 485 nm and an emission wavelength of 530 nm. MDA levels were determined using a thiobarbituric acid-reactive substances assay as previously described with suitable modifications. NO production was quantified spectrophotometrically using Griess reagent (Sigma-Aldrich). The absorbance was measured at 540 nm and nitrite concentration was determined using a calibration curve prepared with sodium nitrite as the standard [36]. Carbonylated protein content was measured according to a previously described method [37].

### 2.9. Measurement of Anti-Oxidative Marker Levels

The expression levels of oxidative and anti-oxidative stress markers were determined as described previously [38]. The levels of anti-oxidative stress markers, glutathione (GSH), catalase (CAT), superoxide dismutase (SOD), glutathione peroxidase (GPx), glutathione S-transferase (GST), and glutathione reductase (GR) were determined according to the manufacturer’s instructions. THP-1 cells were cultured in 75-cm^2^ culture flasks and exposed to IC_50_ concentrations of PtNPs for 24 h. The cells were harvested in chilled PBS by scraping and washing twice with 1× PBS at 4 °C for 6 min each at 1500 rpm. The cell pellet was sonicated at 15 W for 10 s (three cycles) to obtain the cell lysate. The resultant supernatant was stored at −70 °C until analysis.

### 2.10. Measurement of 8-Oxo-7,8-Dihydro-2′-Deoxyguanosine (8-Oxo-Dg) and 8-Oxo-G Levels

8-oxo-dG content was determined as described previously [35,39] and also per the manufacturer’s instructions (Trevigen, Gaithersburg, MD, USA).

### 2.11. Reverse Transcription-Quantitative Polymerase Chain Reaction (RT-qPCR)

Total RNA was extracted from the LNCaP cells treated with various concentrations of PtNPs for 24 h using the PicoPure RNA isolation kit (Arcturus Bioscience, Mountain View, CA, USA). Samples were prepared according to the manufacturer’s instructions. RT-qPCR was conducted using a Vill7 device (Applied Biosystems, Foster City, CA, USA) and SYBR Green as the double-stranded DNA-specific fluorescent dye (Applied Biosystems). Target gene expression levels were normalized to glyceraldehyde-3-phosphate dehydrogenase (*GAPDH*) expression, which was unaffected by the treatment.

### 2.12. Measurement of Cytokine Levels

THP-1 cells were treated with PtNPs for 24 h and cytokine levels were determined using multianalyte inflammatory cytokine ELISA array (Qiagen, Valencia, CA, USA). Sample values were normalized to control values and are displayed as fold change over control.

### 2.13. Statistical Analysis

Independent experiments were repeated at least three times such that data are represented as mean ± standard deviation (SD) for all duplicates within an individual experiment. Data were analyzed using the *t*-test, multivariate analysis, or one-way analysis of variance (ANOVA), followed by Tukey’s test for multiple comparisons to determine the differences between groups denoted by an asterisk. 

## 3. Results and Discussion

### 3.1. Synthesis and Characterization of PtNPs Using Apigenin

In this study, we investigated the cellular responses to PtNPs in THP-1 cells. To demonstrate the cytotoxic, genotoxic, and pro-inflammatory effects of PtNPs, we first confirmed bio-reduction of Pt ions using phenolic compounds such as apigenin as the reducing and stabilizing agent. Bio-reduction was visually verified by initial color change from pale yellow to dark brown, and finally by the appearance of a brownish-black precipitate. The intensity of the color depended on the period of incubation. The color change is due to the reaction between hexachloroplatinic acid and apigenin and indicates the formation of platinum nanoparticles [40]. The entire reaction was completed within 2 h due to the presence of pure phenolic compounds in the solution. The single sharp peak at 300 nm was attributed to the excitations of surface plasmon vibrations, indicating the synthesis of PtNPs (Figure 1A), which is in agreement with the results of previous reports [41,42]. Several studies have reported that metal nanoparticles can be synthesized using biomolecules as reducing and stabilizing agents. For example, silver and copper nanoparticles were prepared using ascorbic acid as the reducing and stabilizing agent [43,44]. In addition to color change, the formation of PtNPs was confirmed by a continuous absorption spectrum in the range of 300–600 nm in a UV-visible spectrophotometer (Figure 1A). The surface plasmon resonance of PtNPs synthesized using apigenin was observed at 302 nm. PtNPs synthesized using Ajwa and Barni dates exhibited resonance at 321 and 329 nm, respectively; however, the absorption spectra are dependent on the size and morphology of the particles.

The purity and crystal structure of the synthesized PtNPs were assessed using XRD. As shown in Figure 1B, the broad reflections of the synthesized nanoparticles indicated the nanocrystalline nature of the powder. Broad diffraction peaks of the XRD pattern were observed at 2θ = 40.6, 48.4, and 68.1, corresponding to reflections (111), (200), and (220), respectively, which are consistent with the face centered cubic (fcc) structure of platinum and can be assigned to JCPDS Card 04-0802, demonstrating the presence of crystalline. Shah et al. [45] reported synthesis of PtNPs with an XRD pattern similar to that reported in this study, using a low-temperature economically feasible method. The sharpening of the peaks clearly indicated that the particles were in the ultra-small size range. The size of the platinum nanocrystals as estimated from the full width at half maximum of the (111) Pt peak using the Scherrer formula was 1–2 nm. The XRD pattern clearly illustrates that the PtNPs formed using this present synthesis method were crystalline in nature.

The bio-reduction and capping of PtNPs was confirmed using FTIR. The typical characteristic bands at 1660 and 3350 cm^−1^ represented the C−N (amide) and O−H bonds, respectively (Figure 1C). The amino acids show the possible interaction of PtNPs and stretching modes for C=O and OH groups. The bio-reduction and capping of PtNPs using water-soluble apigenin was evident from the disappearance of a strong band at 1760 cm^−1^. Furthermore, the presence of a strong band at 1200 cm^−1^ indicated the presence of C−O bond. The binding of biomolecules with platinum nanoparticles via amino acid groups was evident from the shift in NH frequency from 2900 cm^−1^. Al-Radadi et al. [46] demonstrated synthesis of PtNPs using date fruit extracts, which contain polyphenols, organic acids, amino acids, and antioxidants such as apigenin and glutathione, which are responsible for the reduction of Pt salts.

Accurate determination of the size of synthesized nanoparticles is crucial for developing nanoscale technologies and biological studies, as size affects many of the physical and chemical properties of these materials. DLS has been widely used for determining particle size [47]. As shown in Figure 1D, the average size of the particles was 2 nm, which is in accordance with the data obtained from TEM images. The results obtained from DLS exhibited unimodal distribution of nanoparticles with maximum percentage intensity of ~2 nm.

The surface morphology of PtNPs was determined using SEM. As shown in Figure 1E, SEM showed individual PtNPs as well as aggregates. The PtNPs aggregated into large irregular structures with no well-defined morphology and all the particles possessed almost identical morphology. Finally, the structure and size of the PtNPs were analyzed using TEM. The TEM images showed that PtNPs had a uniform spherical shape and regular size, and were well dispersed with a narrow size distribution from 1.0 to 2.0 nm. DLS showed that the particles were 2 nm in diameter (Figure 1F). TEM exhibits significantly smaller size of the particles and good correlation with DLS, as TEM involves imaging of dry nanoparticles, whereas DLS is a measure of the hydrodynamic diameter.

### 3.2. PtNPs Decreased the Survival and Proliferation of THP-1 Cells

First, to evaluate the cytotoxic effects of PtNPs on THP-1 cells, we performed a mitochondria-based viability assay (CCK-8) using different doses of PtNPs. The higher doses of PtNPs (100–150 μg/mL) had a drastic effect on THP-11 viability, reducing cell viability to 60 ± 6.5% after 24 h of exposure (*p* < 0.05). This tendency continued, reaching 90% ± 7% after 24 h (Figure 2A). In contrast, compared to the control cells, low doses of PtNPs (25 µg/mL) did not significantly affect cell viability during the first 24 h of exposure (*p* > 0.05). Viability of THP-1 cells decreased by 90–95% after 24 h of treatment. Comparison of cell viability kinetics of THP-1 indicated that all PtNP treatments reduced the total number of viable cells after 24 h of treatment. Results of the CCK-8 assay showed that PtNPs reduced the number of viable cells in a dose-dependent manner in the concentration range of 25–150 μg/mL.

Next, we evaluated the potential anti-proliferative effect of PtNPs on THP-1 cells. For this purpose, we cultured THP-1 cells in the presence of different concentrations of PtNPs. Treatment of THP-1 with PtNPs (25–150 μg/mL) for 24 h reduced cell proliferation to 25% at 50 μg/mL and to 85% at 150 μg/mL. These observations indicate that PtNPs exhibit dose-dependent cytotoxicity of THP-1 cells by inhibiting cell proliferation (Figure 2B). Yogesh et al. [48] reported in vitro and in vivo activities of biosynthesized PtNPs against human lung cancer, and PtNPs inhibited tumor growth by 66% in Severe combined immunodeficient mice SCID mice. Recently, Gurunathan et al. [38] reported that graphene oxide and reduced graphene oxide reduced the viability and proliferation of THP-1 cells.

### 3.3. PtNPs Altered Cell Morphology

Changes in cell morphology by PtNPs were assessed using a digital microscope (Figure 3). Compared to the control, cells treated with various concentrations of PtNPs displayed significant changes in morphology, including loss of uniformity and remarkable shrinkage around the cell clusters. (Figure 3). The consistency in results obtained for cell viability, cell proliferation, and LDH leakage assay, trypan blue staining, and digital microscopy after PtNP treatments indicated that PtNPs affect cell survival and eventually cause cytotoxicity. Similarly, U87 glioblastoma cells treated with various concentrations of PtNPs showed characteristic cell death morphology with long branched protrusions. Compared to that in the control, PtNPs treatment reduced the cell number and the length of the cell protrusions [49]. At higher concentrations, the cells exposed to PtNPs clearly show membrane blebbing and loss of plasma membrane integrity. Collectively, the results suggest that PtNPs can alter cell morphology. Chronic exposure of human bronchial cells BEAS-2B cells with crystalline silica Min-U-Sil 5 induces epithelial-to-mesenchymal transition via argpyrimidine-modified Hsp70, miR-21 and SMAD signaling [50].

### 3.4. PtNPs Induced Toxicity via Increase in LDH Leakage and Decrease in THP-1 Cell Viability

THP-1 cells were treated with different concentrations (25–150 μg/mL) of PtNPs for 24 h, and LDH leakage, which acts as a sensitive and integrated tool for measuring the integrity of the cell membrane, was determined [35]. Biotoxicity of nanomaterials can be estimated from the leakage of LDH, which is released in culture medium from cells after plasma membrane damage. LDH is commonly known as an indicator of NP penetration into cells; it is ubiquitously present in nearly all cells and is expelled through the damaged plasma membrane into the extracellular space [51]. Plasma membrane damage is the major cause of cell death. Several studies have shown that nanoparticles increase LDH leakage in various cell types, including human breast cancer, ovarian cancer, adenocarcinoma, and neuroblastoma cells [34,47,52,53]. Our results also demonstrated that PtNPs induced LDH leakage in a dose-dependent manner (Figure 4A). Similarly, Wright et al. [54] had demonstrated that carbon nanodots increased LDH leakage in THP-1 cells.

To further confirm the results obtained from the LDH assay, we counted the number of cells after staining with trypan blue. Incubation of cells with various concentrations of PtNPs for 24 h significantly reduced cell viability (Figure 4B). Results suggest that membrane integrity was compromised in PtNPs-treated cells. Kutwin et al. [49] demonstrated the effect of PtNPs on U87 glioma cells; their results suggested that the cells treated with PtNPs showed morphological deformations, DNA damage, decreased metabolic activity, and genotoxic effects. Exposure of human cervical cancer SiHa cells to platinum-copper alloy nanoparticles inhibited cell proliferation and enhanced nuclear morphological changes, including cell shrinkage, intranucleosomal DNA fragmentation, and chromatin condensation [55]. Collectively, PtNPs potentially induce membrane damage in THP-1 cells. Similarly, carbon nanoparticles such as graphene oxide and reduced graphene oxide increased LDH level and decreased THP-1 proliferation [38].

### 3.5. PtNPs Induced ROS Generation and Expression of Oxidative Stress Markers

Oxidative stress results from an imbalance between ROS production and ROS scavenging activity in cells. The balance between pro- and anti-oxidants determines various cellular functions, signal transductions pathways, and cellular metabolism [56]. ROS play an important role in the progression of several diseases, including inflammation, atherosclerosis, aging, and age-related degenerative disorders. We determined the effect of PtNPs on ROS induction in THP-1 cells. PtNPs induced the generation of significant levels of ROS in THP-1 cells (Figure 5A). PtNPs modulate oxidative stress by impairing receptor activator of nuclear factor-κB ligand (RANKL) signaling [57]. Our results showed that ROS production is a common mechanism of toxicity both in vitro and in vivo.

Furthermore, oxidative stress induced by PtNPs has been correlated with lipid peroxidation (Figure 5B), which increases MDA levels in cells, eventually leading to oxidative damage of mitochondrial DNA, alterations in the mitochondrial membrane potential, and changes in the expression of key antioxidant enzymes such as SOD2 [58,59,60,61]. The exposure of THP-1 cells to PtNPs increased MDA concentration in a dose-dependent manner after 24 h. As shown in Figure 5B, compared to that in the control, MDA level increased continuously with PtNP dose. Our results suggest that adaptation of the antioxidative system was not sufficient to prevent damage to membrane lipids induced by increasing doses of PtNPs. Significant elevation of MDA levels was observed in PtNP-treated THP-1 cells. The lipid peroxidation (LPO) products may subsequently affect the structure of DNA bases [62], proteins, and carbohydrates. Similar results were reported in Medical Research Council cell MRC [63] and human bronchial epithelial BEAS-2B cells exposed to TiO_2_ nanoparticles [58] and silver nanoparticles [35]. LPO-generated 4-hydroxynonenal (HNE) and its protein adducts as possible mechanism of PtNPs induced cytotoxicity in THP-1 cells. Although we measured LPO levels using TABARS, this assay cannot be considered an optimal bio-analysis method for LPO. Therefore, the future focus will be on immunochemical analysis of HNE-protein adducts, which is a more reliable and significant biomarker and could further support mechanistic studies on PtNP bioactivities. To further investigate the PtNP-induced ROS-mediated cellular responses, we determined the levels of reactive nitrogen species such as the NO radical and products that are formed by its reaction with the superoxide anions, e.g., peroxynitrite and nitrous oxide radicals [64]. NO is one of the most widespread signaling molecules in mammals. It modulates various physiological reactions, including vasodilation and relaxation of smooth muscles associated with circulation, neurotransmission in various neural processes, and regulation of immunological defense mechanisms [65]. The effect of PtNPs on NO production and inflammatory responses remains unclear. Cells treated with PtNPs showed higher levels of NO than the control groups (Figure 5C). The results obtained from the NO assay are consistent with those obtained from the cell viability and LDH assays. Exposure of airway epithelial cells to crystalline silica induces mitochondrial mediated apoptotic pathway via a novel mechanism involving Hsp70, JNK, and NF-κB [66].

Protein carbonylation, a marker of oxidative stress, is involved in various metabolic diseases. Carbonylation is dependent upon the severity of oxidative stress in cells. For instance, in highly oxidative cells such as macrophages, ROS produced by the NADPH oxidase system may lead to the formation of significant amounts of lipid aldehydes that covalently modify endogenous enzymes and proteins [67]. Therefore, we determined the dose-dependent effect of PtNPs on carbonylation of proteins in THP-cells. Results showed that increasing the dose of PtNPs directly increased intracellular protein carbonyl content by 10–18-fold compared to control cells (Figure 5D). The results regarding these oxidative stress biomarkers indicate a close relationship between cytotoxicity and oxidative stress generated by PtNP treatment. Antognelli et al. (2009) [68] demonstrated the effect of crystalline silica Min-U-Sil 5 on oxidative stress, efficiency of antiglycation and antioxidant enzymatic defenses.

Similarly, after treatment with silver nanoparticles, THP-1-derived macrophages showed an approximately 3-fold higher level of protein carbonyls than control cells [69]. Thus, we concluded that PtNP-induced oxidative stress is responsible for the observed cytotoxic effects. Recently, Gurunathan et al. (2019) [38] reported that graphene oxide and reduced graphene oxide increased the level of oxidative stress in THP-1 cells by increasing ROS levels, lipid peroxidation, nitric oxide levels, and protein carbonylation.

### 3.6. Effect of PtNPs on Antioxidant Markers

The most common negative outcome of the use of nanoparticles as therapeutic molecules for cancer is the excessive generation of ROS, which is a key factor in NP-induced toxicity [70]. Imbalance in oxidative status occurs due to production of higher levels of ROS in the system. Thus, antioxidants are required to scavenge the excess free radicals. Reoccurrence of oxidative stress eventually leads to inflammation. Studies have reported that PtNPs are suitable agents for reducing ROS levels under certain conditions [71]. Hence, we attempted to determine the effect of higher concentration and smaller size (larger surface area) of PtNPs on THP-1 cells. To determine the effect of PtNP size on the cellular levels of various antioxidants, we analyzed the expression of markers such as GSH, CAT, SOD, GPx, GST, and GR. Results revealed that the levels of all the tested markers were significantly lower in the treated cells than in the control, and the reduction occurred in a dose-dependent manner (Figure 6).

The potential effect of different metallic nanoparticles on the production of either ROS or antioxidants depends on their chemical composition, particle size, surface area, and shape, as well as on the mode of interaction with cells, aggregation, inflammation, and pH of the medium [70,72]. Importantly, the imbalance between ROS and antioxidants at the cellular level is dependent on the concentration of NPs. Exposure to low NP concentration improves the endogenous antioxidant defense system, which combats the damaging consequences of oxidative stress and recovers redox balance, whereas exposure to high NP concentrations induces excess ROS formation, which overwhelms the antioxidant system and causes cytotoxicity and inflammation [73]. Similarly, the exposure of PC12 cells to SiO_2_-NPs for 24 h resulted in apoptosis, accompanied by increased intracellular ROS levels and oxidative damage due to depletion of glutathione (GSH), production of methane dicarboxylic aldehyde, and inhibition of SOD. Accordingly, the antioxidant defense system of THP-1 cells reacted negatively to PtNP toxicity by decreasing the level of antioxidant markers. The possible mechanism underlying reduction in the level of antioxidants involved interaction between the sulfhydryl groups or inhibition of several cellular activities involving cysteine uptake [63]. Collectively, our results indicated that reductions in the levels of GSH, CAT, SOD, GPx, GST, and GR, and elevations of the levels of ROS, MDA and carbonylated protein, impede the activity of the antioxidant defense mechanisms in THP-1 cells, which cannot efficiently counteract the oxidative stress induced by exposure to PtNPs, resulting in cellular damage. Recently, Gurunathan et al. (2019) [38] reported that graphene oxide and reduced graphene oxide treatment lowered the level of antioxidant proteins in THP-1 cells.

### 3.7. PtNPs Impaired Mitochondrial Membrane Potential (MMP) and Decreased ATP Level

Mitochondria are vital organelles involved in the nanoparticle-induced generation of cellular ROS via depolarization of MMP and interference with the electron-transport chain [73]. NP exposure can block the mitochondrial electron-transport chain, consequently increasing the cellular levels of superoxide radicals via electron transfer from respiratory carriers to molecular oxygen [74]. To determine the effect of PtNPs on MMP in THP-1 cells, the cells were treated with various concentrations of PtNPs for 24 h and the MMP level was determined. The MMP level in cells treated with various concentrations of PtNPs was lower than that in the control group (Figure 7A). Almeer et al. (2018) [75] suggested that a decrease in MMP level is cytotoxic to HEK293 cells.

The ATP level is crucial for regulating cellular metabolic activity; therefore, we investigated the dose-dependent effect of PtNP on ATP level. The results showed that ATP content decreased in cells treated with PtNPs, which is in agreement with the decrease in cellular metabolism in the cell viability test (Figure 7B). Similarly, the exposure of human colon cancer cells to silver nanoparticles altered MMP, lowered ATP concentration, and increased Bax expression. Recently, Gurunathan et al. (2019) [38] reported that graphene oxide and reduced graphene oxide treatment lowers ATP level in THP-1 cells.

### 3.8. PtNPs Increased the Expression of Proapoptotic Genes and Decreased the Expression of Antiapoptotic Genes

To investigate the mechanisms associated with DNA damage and apoptosis in PtNPs-treated THP-1 cells, we determined the expression of key genes involved in mitochondria-mediated apoptosis. Results showed significantly higher expression of proapoptotic genes such as those encoding p53, p21, Bax, Bak, caspase 9, and caspase 3 (Figure 8). A previous study showed that exposure of THP-1 cells to ZnONPs increased the expression of endoplasmic reticulum (ER) stress- and apoptosis-related gene expression, which is dependent on the interactions between ZnONPs and biological molecules [76]. Similarly, Almeer et al. [75] reported increase in the levels of Bax and caspase 3 and decrease in the level of Bcl2 in PtNP-treated HEK293 cells. The downregulation of Bcl-2 mRNA and concomitant upregulation of proapoptotic genes indicated that Bcl-2 and Bax play a key role in the execution of apoptosis and that the Bcl-2 family is involved in the regulation of apoptosis. Similarly, carbon nanoparticles such as graphene oxide and reduced graphene oxide increased the expression of proapoptotic genes and downregulated antiapoptotic genes in THP-1 cells.

### 3.9. PtNPs Increased Oxidative Damage to DNA and RNA

Nanoparticle-induced oxidative stress and DNA damage may alter cell proliferation, differentiation, or cell-to-cell signaling [77]. It is well-known that nanoparticle-induced ROS can generate 8-oxodG and 8-oxo-G, markers of oxidative DNA and RNA damage, respectively [78], and major ROS-induced oxidative modifiers [79,80]. 8-Oxo-dG and 8-oxo-G levels were determined to evaluate the effects of PtNP-induced ROS on DNA damage in THP-1 cells. Exposure of THP-1 cells to various concentrations of PtNPs for 24 h increased oxidative DNA damage, as indicated by a significant elevation in 8-oxo-dG and 8-oxo-G production. Results indicated that PtNPs dose-dependently increased the level of 8-oxo-dG and 8-oxo-G (Figure 9A,B). Compared to the control groups, a significant difference was observed in DNA and RNA damage by measuring the levels of 8-oxo-dG and 8-oxo-G after exposure to PtNPs for 24 h. Hydroxyl radical is a crucial molecule that damages nucleic acids and other biomolecules. Excessive hydroxyl radical attacks adjacent DNA strands both in the nucleus and mitochondria, eventually generating different types of oxidation products [81]. RNA is more susceptible than DNA due to lack of specific RNA-protecting proteins in living cells [82]. Gurunathan and colleagues have observed that carbon nanomaterials such as graphene oxide and reduced graphene oxide cause oxidative DNA damage by increasing the levels of 8-oxodG and 8-oxo-G in THP-1 cells [38]. The mechanism underlying DNA damage might involve cross-linking of phospholipid membranes by smaller PtNPs, which reach the nucleus and bind to DNA [83]. This might lead to direct induction of DNA damage by these small particles. Collectively, the smaller PtNPs (2 nm) can potentially cause DNA damage in THP-1 cells.

### 3.10. PtNPs Regulated DNA Damage and Repair Genes

Oxidative stress is one of the most common mechanisms of nanoparticle-induced DNA damage, and oxidatively-induced DNA lesions are predominantly repaired by the base excision repair (BER) pathway. Damage to DNA bases due to oxidative stress are deleterious, leading to stalled replication forks, mutations, and eventually cell death [84]. To understand the genotoxicity of PtNPs, we assessed the expression of various genes involved in DNA repair as an adaptive response to PtNPs. We determined the expression levels of relevant DNA glycosylases involved in BER, such as *OGG1, APEX1, CREB1, POLB, UMG,* and GADD45A. PtNP treatment upregulated all the tested genes in a dose-dependent manner (Figure 10). These results indicated that the PtNPs modulate the expression of DNA glycosylases. Titanium dioxide NPs (TiO2 NPs) induce the activation of the serine/threonine kinase ATM/Chk2, which is involved in the DDS signaling pathway [85]. TiO2 NPs-induced increase in the expression of ATM in hepatocellular carcinoma cells (HepG2) leads to induction of double strand breaks (DSBs), chromatin condensation, nuclear fragmentation, and apoptosis due to increased ROS production and subsequent DNA damage [86]. The exposure of cells to engineered nanomaterials affects the functioning of the entire cellular system, including fidelity of DNA replication and cell division, resulting in wide ranging DNA lesions that include genome rearrangements, single strand breaks (SSBs), DSBs, intra/inter strand breaks (SBs), and formation of modified bases. DNA lesions can lead to chromosomal aberrations, mutations, apoptosis, carcinogenesis, or cellular senescence [87,88,89,90]. Gurunathan and colleagues observed that carbon nanomaterials such as graphene oxide and reduced graphene oxide cause oxidative DNA damage by increasing the expression of DNA damage and repair genes in THP-1 cells [38].

### 3.11. PtNPs Increased the Expression of Proinflammatory Cytokines and Chemokines

ROS cause oxidative damage to various macromolecules, including proteins, lipids, and nucleic acids [91]. Repair of DNA lesions is crucial for maintaining genomic integrity, whereas other oxidatively damaged macromolecules undergo degradation [92]. Previous studies have reported that proinflammatory cytokines are induced after exposure of RAW264.7 macrophages [93] and THP-1 monocytes [94,95] to PM2.5. However, studies on the effect of PtNPs on THP-1 cells are lacking so far. Therefore, we investigated the molecular mechanism of macrophage inflammatory responses to PtNPs. To determine whether proinflammatory cytokines are upregulated in THP-1 cells after exposure to PtNPs, we determined the protein levels of interleukin-1β (IL-1β), IL-6, IL-8, tumor necrosis factor-α (TNF-α), granulocyte-macrophage colony-stimulating factor (GM-CSF), and monocyte chemoattractant protein 1 (MCP-1). Exposure of THP-1 cells to PtNPs significantly increased the expression levels of IL-1β, IL-6, IL-8, TNF-α, GM-CSF, and MCP-1 in a dose-dependent manner (Figure 11).

Gold nanoparticles (GNPs) altered proinflammatory cytokine expression in rat liver and kidneys. Exposure of innate immune cells such as macrophages, dendritic cells, and monocytes to SiNPs increased production of pro-inflammatory cytokines such as IL-1b and IL-18 [96]. The expression levels of IL-1, IL-6, and TNF-α were significantly increased in silver nanoparticle-treated THP-1 cells and primary blood monocytes [97]. Recently, Khan et al. [98] reported that a single intraperitoneal injection of small-sized GNPs (5 nm) significantly increased IL-1 beta and IL-6 mRNA expressions in mouse brain, whereas the larger GNPs (20 and 50 nm) did not produce any inflammatory response. This indicates that the size of the nanoparticles plays a critical role in cytokine production. Gomez et al. [99] evaluated the potential in vitro immunomodulatory effect of 12-nm and 200-nm SiNPs on the expression of pro-inflammatory cytokines and NLRP3 inflammasome components in human primary neutrophils and peripheral blood mononuclear cell PBMCs. They observed that SiNPs induced the production of pro-inflammatory cytokines in a dose-dependent manner. Exposure of macrophages to atmospheric particulate matter with aerodynamic diameter less than 2.5 μm (PM2.5) significantly increased the intracellular levels of TNF-α, IL-1β, and IL-6, in both a dose- and time-dependent manner. Recently, Gurunathan et al. [38] suggests that graphene oxide and reduced graphene oxide aberrantly increase the expression of proinflammatory cytokines and chemokines. Our findings demonstrate that the smaller size of PtNPs is responsible for the production of proinflammatory cytokines in a dose-dependent manner.

## 4. Conclusions

Owing to their unique physical and chemical properties, metallic nanoparticles (NPs) are being used in biomedical applications and as consumer products. Pt is also used as a vehicle exhaust catalyst, leading to possible exposure via inhalation. Despite their use, data on the cytotoxic, genotoxic, and proinflammatory responses to them and possible size-dependent effects are limited, particularly for PtNPs. Therefore, we analyzed the effect of ultra-small PtNPs (1–2 nm) on cytotoxicity, genotoxicity, and proinflammatory responses in THP-1 cells. First, we synthesized PtNPs using apigenin and characterized them using various analytical techniques. PtNPs caused cytotoxicity by decreasing cell viability and proliferation in a dose-dependent manner. The potential cytotoxicity of PtNPs on cellular redox systems was determined by determining LDH leakage, ROS generation, lipid peroxidation, and expression of antioxidants. Interestingly, LPO-generated 4-hydroxynonenal (HNE) and its protein adducts could play crucial role in cytoto, geno and immunotoxicity and these products could be a possible mechanism of PtNPs-induced toxicity (Figure 12). Studies are required to address these biological effects of PtNP which will be a focus point of future research. The results suggest that PtNPs potentially induce oxidative stress by increasing the levels of oxidative stress markers such as ROS and lipid peroxidation, and decrease the levels of antioxidant markers. The involvement of mitochondria in oxidative stress and genotoxicity was determined by assessing MMP, ATP level, and gene expression. The results of all these cellular assays clearly suggest that PtNPs treatment potentially leads to loss of mitochondrial functions. The possible mechanism of PtNPs-induced cell death was confirmed by gene expression analysis, which revealed upregulation of apoptotic markers such as p53, p21, Bax, Bak, caspase 9, and caspase 3, and simultaneous downregulation of antiapoptotic genes encoding Bcl-2 and Bcl-xl. The impact on DNA damage was confirmed by increased expression of DNA damage and repair genes associated with increases in the levels of 8-oxoG. PtNP treatment increased oxidative DNA damage and impaired DNA integrity. Altogether, PtNPs were able to stimulate multiple stress responses such as oxidative stress, cytotoxicity, genotoxicity, and secretion of proinflammatory cytokines. However, further in vivo and molecular mechanistic studies are necessary to understand the trafficking patterns of these nanoparticles inside cells and the mechanism underlying the proinflammatory response.

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
