# Peer review of "The Effects of Apigenin-Biosynthesized Ultra-Small Platinum Nanoparticles on the Human Monocytic THP-1 Cell Line"

_cells, 2019, doi:10.3390/cells8050444_

Reviewer 1 Report

The authors report on preparation of platinum nanoparticles by reduction of platinum(IV) by apigenin. The particles were characterized by many techniques like DLS, SEM, TEM, X-ray, IR, and UV-Vis spectroscopies. Cell viability and cell proliferation were studied. Oxidation stress was determined as well as DNA damage by increase of 8-oxo-7,8-dihydro-2′-deoxyguanosine.

The manuscript is well written and the study is very well documented. The abstract seems to be too long. Shortcuts 8-oxodG and 8-oxoG are not explained in abstract. There is a question why SEM figure was shown. What is in the figure?

Author Response

Response to the reviewer comments-1

We immensely thank the reviewer valuable and constructive comments that greatly facilitated us for improving the overall quality of the manuscript. As per the reviewers’ constructive comments, the corrections were carried out in the manuscript. We hopefully believe that we have addressed all the comments mentioned by the reviewers carefully and precisely. All the changes are highlighted in yellow color in the revised manuscript. In addition, this manuscript was proof read by native English speaker by Editage editing company, Seoul, South Korea.

Comments and Suggestions for Authors

The authors report on preparation of platinum nanoparticles by reduction of platinum(IV) by apigenin. The particles were characterized by many techniques like DLS, SEM, TEM, X-ray, IR, and UV-Vis spectroscopies. Cell viability and cell proliferation were studied. Oxidation stress was determined as well as DNA damage by increase of 8-oxo-7,8-dihydro-2′-deoxyguanosine.

Thanks to the reviewer for encouraging, positive and constructive comments to improve overall quality of the manuscript.

The manuscript is well written and the study is very well documented. The abstract seems to be too long. Shortcuts 8-oxodG and 8-oxoG are not explained in abstract. There is a question why SEM figure was shown. What is in the figure?

Once again thanks to the reviewer for encouraging comments about our work and presentation of results.

Response to your first comment, we reduced the length of the abstract; we reduced the word count from 320 to 296.  The length of the abstract is due to extensive data presented in the manuscript.

Response to second comments, we included the oxidative DNA damage due to increased level of 8-oxodG and 8-oxoG in the abstract.

Response to your third comment, SEM figure was included to show the surface morphology of synthesized particles.

Once again thanks to the reviewer for wonderful comments to improve the overall quality of the manuscript.

Reviewer 2 Report

The study by Sangiliyandi Gurunathan and colleagues describes the cytotoxic, gentotoxic and pro-inflammatory effects of Platinum nanoparticles on THP-1 cells. The study is original and current, and methods are appropriate. Results are clearly described and discussed in the context of the literature.

Nevertheless, the study is merely descriptive and no mechanisms have been provided even though it reports a large amount of results related to cytotoxicity, genotoxicity and pro-inflammatory aspects.

Specific comments:

1) the work would have benefited of an additional cell line where investigate the biological processes here investigated; unless there is a specific reason, in which case please explain why you chose THP-1.

2) the doses of PtNPs you used were of physiological importance? How did you choose them?

3) TEM imaging would have improved the work.

4) Introduction: some parts are repetitive, please, change.

5) I would have improved the study by performing a chronic exposure to low doses of PtNPs and evaluated at least some of the biological phoenomena studied in the acute exposure. Please, comment this in Discussion. Citing a significant and very impressive recent article could be helpful in such a sense (Free Radic Biol Med. 2016;92:110-125. doi: 10.1016/j.freeradbiomed.2016.01.009).

6) I would add in the appropriate parts of the Discussion the following references showing the interplay between oxidative stress and antioxidant enzymatic defenses  and/or with apoptosis (Chem Biol Interact. 2009;182(1):13-21. doi: 10.1016/j.cbi.2009.08.002 and Free Radic Biol Med. 2015;84:128-141. doi: 10.1016/j.freeradbiomed.2015.03.026), in support of your results.

7) English requires a minor revision.

Author Response

Response to the reviewer comments-2

We immensely thank the reviewer valuable and constructive comments that greatly facilitated us for improving the overall quality of the manuscript. As per the reviewers’ constructive comments, the corrections were carried out in the manuscript. We hopefully believe that we have addressed all the comments mentioned by the reviewers carefully and precisely. All the changes are highlighted in yellow color in the revised manuscript. In addition, this manuscript was proof read by native English speaker by Editage editing company, Seoul, South Korea.

Comments and Suggestions for Authors

The study by Sangiliyandi Gurunathan and colleagues describes the cytotoxic, gentotoxic and pro-inflammatory effects of Platinum nanoparticles on THP-1 cells. The study is original and current, and methods are appropriate. Results are clearly described and discussed in the context of the literature. Nevertheless, the study is merely descriptive and no mechanisms have been provided even though it reports a large amount of results related to cytotoxicity, genotoxicity and pro-inflammatory aspects.

Thanks to the reviewer for encouraging, positive and constructive comments to improve overall quality of the manuscript.

Specific comments:

1) the work would have benefited of an additional cell line where investigate the biological processes here investigated; unless there is a specific reason, in which case please explain why you chose THP-1.

THP-1 designates a spontaneously immortalized monocyte-like cell line, derived from the peripheral blood of a childhood case of acute monocytic leukemia (M5 subtype) (3). THP-1 cells, including their genetically engineered derivatives, represent valuable tools for investigating monocyte structure and function in both health and disease. The use of cultured THP-1 cells in vitro as a model for primary human monocytes ex vivo exemplifies the basic concept of translational research. (Herbert Bosshartand Michael Heinzelmann, 2016).

2) the doses of PtNPs you used were of physiological importance? How did you choose them?

Thanks to the reviewer for excellent and thought-provoking question. Based on the literature available, the current usage of cisplatin for the cancer patient is approximately between 50 and 150 mg. We used for our current study is maximum 150 microgram for in vitro cell culture. The used dose for our study is very low dose, definitely the used dose is comparable. Therefore, we selected the dose from 25-150 microgram/mL.

3) TEM imaging would have improved the work.

Thanks to the reviewer for the comments to increase the quality of the manuscript. The editor has given only one week to resubmit the revised work based on other reviewers comments. Therefore within the stipulated time, including TEM images are seems to be difficult. Anyway we will consider these valuable comments in our future work.

4) Introduction: some parts are repetitive, please, change.

Thanks to the reviewer for critical evaluation. Yes, some words like endotoxin, time consuming and etc was found repetitive. We deleted all repetitive words in the abstract.

5) I would have improved the study by performing a chronic exposure to low doses of PtNPs and evaluated at least some of the biological phoenomena studied in the acute exposure. Please, comment this in Discussion. Citing a significant and very impressive recent article could be helpful in such a sense (Free Radic Biol Med. 2016;92:110-125. doi: 10.1016/j.freeradbiomed.2016.01.009).

Thanks to the reviewer for excellent idea. We performed with low doses from 5-20 µg/mL, but the data indicate there is no significant. Therefore we didn’t include in the manuscript, and also we performed short exposure with high doses of PtNPs with 200 µg/mL, the obtained results similar with 150  µg/mL. Therefore, finally we performed and presented dose-dependent effect of PtNPs.

Response to your second comment, we cited and discussed the manuscript in the revised manuscript.

6) I would add in the appropriate parts of the Discussion the following references showing the interplay between oxidative stress and antioxidant enzymatic defenses  and/or with apoptosis (‍‍Chem Biol Interact. 2009;182(1):13-21. doi: 10.1016/j.cbi.2009.08.002 and Fre‍‍e Radic Biol Med. 2015;84:128-141. doi: 10.1016/j.freeradbiomed.2015.03.026), in support of your results.

Thanks to the reviewer to include supporting literature. According to the reviewer suggestions we cited and discussed both references in the references.

7) English requires a minor revision.

This revised manuscript was once again proof read by native English speaker by Editage editing company, Seoul, South Korea.

Once again thanks to the reviewer for wonderful comments to improve the overall quality of the manuscript.

Round  2

Reviewer 2 Report

Authors have adequately responded to some of my comments and provided convincing justification to other comments. I believe the manuscript has been significantly 
improved and now warrants publication in Cells.